# Lessons learned from cross-sectoral collaboration to protect migrant farmworkers during COVID-19 in Spain

**Miquel Úbeda**[1⚬], **Vanesa Villa-Cordero**[2⚬], **Agustín González-Rodríguez**[1,3], **Sergio Andrés-Cabello**[4], **Iratxe Perez-Urdiales**[5,6*], **María del Mar Jiménez-Lasserrotte**[7], **Mar Pastor-Bravo**[8], **Erica Briones-Vozmediano**[1,9,10]

1 Department and Faculty of Nursing and Physiotherapy, University of Lleida, Lleida, Spain, 2 Department of Health Sciences, Gimbernat University Schools, Barcelona, Spain, 3 Grup de Recerca Anàlisi Social i Educativa (GR-ASE), University of Lleida, Lleida, Spain, 4 Department of Human Sciences, University of La Rioja, La Rioja, Spain, 5 Department of Nursing I, University of the Basque Country (UPV/EHU), Biscay, Spain, 6 Biocruces Bizkaia Health Research Institute, Biscay, Spain, 7 Department of Nursing, Physiotherapy, and Medicine, University of Almería, Almería, Spain, 8 Department of Nursing, University of Murcia, Cartagena (Murcia), Spain, 9 Consolidated Research Group in Society, Health, Education and Culture (GESEC), Lleida, Spain, 10 Research Group in Healthcare (GRECS), Biomedical Research Institute (IRB), Lleida, Spain

⚬ These authors contributed equally to this work.
* iratxe.perez@ehu.eus

**Citation:** Úbeda M, Villa-Cordero V, González-Rodríguez A, Andrés-Cabello S, Perez-Urdiales I, Jiménez-Lasserrotte MdM, et al. (2025) Lessons learned from cross-sectoral collaboration to protect migrant farmworkers during COVID-19 in Spain. PLoS ONE 20(1): e0307578. https://doi.org/10.1371/journal.pone.0307578

**Data Availability Statement:** The data that support the findings of this study contain potentially identifying participant information, so data will be

## Abstract

In Spain, the agricultural sector relies heavily on migrant workers, especially during seasonal seasons. However, these workers face significant challenges related to precarious working conditions and structural vulnerability, which have become more acute since the outbreak of the COVID-19 pandemic. This descriptive qualitative study was based on 87 personal interviews with health and social professionals from sectors such as NGOs, social services, trade unions, local institutions, and health services to promote compliance with these measures in four Spanish regions. It explored the difficulties faced by migrant agricultural workers in complying with prevention measures during the COVID-19 pandemic, the measures taken by organizations and public institutions and the health consequences of the pandemic on this group. After a reflexive thematic analysis, the results show how to mitigate the pandemic's impact, both local authorities and NGOs implemented extraordinary measures to care for migrant farm workers. Responses to the pandemic included temporary housing, mass testing, and stricter labor regulations. Working conditions, constant mobility, precarious housing, and language barriers contributed to the difficulty of implementing preventive measures. Covid-19 intensified inequalities and highlighted the lack of preparedness of institutions to deal with this group. The conclusions suggest that intercultural competence in health professional training and the creation of inclusive approaches to health and social care are critical to addressing health disparities and ensuring the well-being of all migrant farm workers, regardless of their migration status or mobility.

available from the Biomedical Research Institute of Lleida (contact via info@irblleida.cat) upon resonable request.

**Funding:** The study was supported by the Instituto de Salud Carlos III (ISCIII), under the grant: PI20/01310 awarded to EBV. The funder had no role in study design, data collection and analysis, decision to publish, or preparation of the manuscript.

**Competing interests:** The authors have declared that no competing interests exist.

## Introduction

Spain is known for its intensive agriculture, providing fruit and vegetables to many of Europe [1]. Its geographical, economic, and social situation is also an attractive factor for foreign nationals wishing to emigrate. In 2022, 7.534.513 foreign-born persons were registered in Spain [2]. Of these, around 1.5 million citizens work in the agricultural sector, where in orchards, greenhouses, nurseries and gardens, migrant workers represent 66.83% of those hired in the sector with the professional category of labourer, and 36.63% as skilled workers. Not in vain, in 2022, the arrival in Spain of 31.219 irregular immigrants was recorded, of which almost 93% of them arrived by sea and by means of precarious boats dedicated to the illegal trafficking of people. By country of origin, the profile of irregular immigrants in Spain corresponds to citizens from Morocco, Guinea, Algeria, Mali, Côte d'Ivoire, Senegal and Tunisia [3, 4] and 1.258.894 regular immigrants, the main nationalities being Colombian (with 172.241 arrivals in Spain), Moroccan (113.725) and Ukrainian (91.621) [5].

The transformations in the sector in recent decades have led to a greater dependence on MSWs. As a result of the demands of the labor activity and the low wages in the tasks of harvesting and maintenance of crops, since the 1980s, there has been a gradual change where the native population has left the sector and has been replaced by workers of migrant origin [6]. Thus, throughout the year, different agricultural regions concentrate thousands of workers in search of opportunities that they cannot find in their countries of origin [7, 8]. By origin, the usual profile of agricultural migrant workers is that of citizens of African (Maghreb) and Eastern European (Bulgaria and Romania) origin [9]. This collective became, therefore, a determining factor for many crops, such as fruit trees and vineyards, and work in greenhouses, for example [10].

MFWs could be contracted in advance in their origin countries due to international political agreements (specially women from Morocco in the south of the country, and Latin American and East Europe workers), or *in situ* spontaneously (which is more common for African men). The employment contract of migrant agricultural workers is usually temporary. Even if MFW characteristics, origins and their working situations differ between regions in Spain, around 58% of them move between provinces for temporary jobs during the different harvesting seasons [11].

This change in the Spanish agricultural labor composition, characterized by the intense use of migrant workers, is directly related to labor precariousness and places these workers in a position susceptible to abuse or exploitation, especially when these workers are in an irregular situation [12, 13]. Over the last decades, the Administrations have been increasing measures to improve these conditions, as well as controls over employers to avoid irregularities and abuses and to ensure that their rights are complied with [14]. The concerns related to labor rights, vulnerability, and social exclusion of these workers have been documented by various national and European sources [15, 16], causing the agri-food sector to currently face several complex and multifaceted challenges to ensure that agricultural products are produced in conditions of safety and dignity for workers [17]. These precarious living conditions have a direct impact on people's health and quality of life [18–20]. Precarious employment implies greater mobility that conditions migrant seasonal workers (MSW) access to decent housing or the health system [21, 22]. It also exposes MSW to a higher risk of injury from occupational accidents [23]. Thus, most of the health problems of MSW detected by different studies are associated with socioeconomic inequality, with a high prevalence of musculoskeletal problems derived from working conditions, non-communicable diseases such as diabetes mellitus, cardiac and renal alterations [18, 24, 25], in addition to communicable diseases prevalent in their places of origin [26, 27].

Although, in absolute data, immigration is equally distributed between men and women (48% men and 52% women), the gender gap among agricultural migrant workers is significant, with women accounting for only 25% of all hires [11]. Women MSW in southern Spain follow a general trend of feminization of migrations motivated by the renationalization of the global care chain [28]. Most women arrive with contracts from their countries of origin to work in intensive agriculture. The working conditions they found upon arrival are marked by exploitation and situations of labor and sexual harassment [29]. Regarding living conditions, sub-Saharan women usually live in farmhouses and greenhouses, sharing small spaces with other women in groups of 4 to 10 people. They share rooms with beds separated by curtains. MSW women of Moroccan origin, and mainly seasonal workers from Huelva, often stay in Spain, encouraged by false promises of work and residence (migrants nearby). Without resources or a support network, MSW women live poorly in settlements where sanitary conditions are minimal and often resort to prostitution in the settlement itself [30, 31].

The COVID-19 pandemic situation increased the health vulnerability of migrants. In Spain, mobility was limited except for essential activities, such as work in intensive agriculture. Due to unequal employment, housing, and living conditions that migrant workers usually face, complying with prevention measures to limit COVID-19 transmission was complicated, including continuous hand washing, mandatory use of masks, and home confinement [24, 32–34]. Consequently, the risk of disease transmission during the pathogenic period was higher in the migrant population than in the native population [35–38]. Different studies provide epidemiological data on the impact of COVID-19 on virus transmission in the migrant population [39], on living conditions in settlements [16], or inequalities in care, safety, and health of agricultural workers [40–42]. These studies offer epidemiological and social perspectives on the impact of COVID-19, addressing issues such as the spread of the virus, living conditions in settlements, and disparities in care and occupational safety, thus providing a comprehensive view of the challenges faced by agricultural workers in different contexts. The agri-food sector was considered strategic per article 15 Royal Decree 463/2020, which declares a state of alarm and establishes the necessary measures to guarantee food supply throughout the national territory [43].

The existing support landscape for MSWs in Spain before the pandemic was characterized by structural challenges, including limited access to healthcare, inadequate housing, and the complexities of navigating legal and social services, often exacerbated by language barriers and uncertain immigration status. During the pandemic, both local authorities and NGOs implemented extraordinary measures to care for MSWs, including temporary housing to ensure social distancing, mass testing, and stricter labor regulations to enhance safety in the workplace. Health education initiatives were carried out by NGOs, often in collaboration with agricultural companies, to inform MSWs about prevention measures and facilitate access to healthcare services [40, 44, 45].

However, the support landscape had its challenges. The inability to stop working, poor housing conditions, and cultural and linguistic differences hampered adherence to primary and secondary prevention measures, impacting the health of seasonal workers and their access to healthcare services. Additionally, the precarious nature of MSWs' employment and housing conditions contributed to the difficulty of implementing preventive measures effectively [21, 46].

Additional deeper analysis of the perspective of professionals from various sectors involved in caring for this vulnerable population, such as NGOs, social services, trade unions, local institutions, and health services, would help to understand the challenges inherent in providing services to this group [47]. To address this gap, this study delves both into the difficulties experienced by MSW in complying with prevention measures during the COVID-19 pandemic, the health consequences of the pandemic on them and the actions undertaken to

promote compliance with these measures in four Spanish regions that host high numbers of MSWs (Catalonia, La Rioja, Murcia, and Andalusia) from the health and social professionals' perspective.

## Materials and methods

### Study design

The study is part of a larger mix-methods research project that aims to assess the impact of job insecurity and social exclusion on the health of migrant men and women working in the agricultural sector in Spain [48]. This qualitative study focuses on the experiences shared by social and health professionals during the COVID-19 pandemic in four specific regions of Spain: Catalonia, La Rioja, Murcia, and Andalusia.

We employed a descriptive qualitative approach that allows researchers, from a naturalistic perspective, to describe little-known phenomena and interpret participants' experiences in their everyday environments [49]. This method allowed us to explore and understand how the COVID-19 pandemic affected MSW, focusing on the perceptions of those involved in the migrants' care in a social and healthcare context.

### Participants and data collection

Six of the authors of this article, three men and three women conducted the interviews. We conducted 87 semi-structured personal and online interviews (80 individual and 6 in pair interviews) with 92 participants (55 women and 37 men) between 24th October 2021 and 8th July 2022. The interviews involved professionals from public and social entities involved assisting MSWs: NGO staff, health workers, social workers and educators, intercultural mediators, agricultural trade unions, and people with managerial positions in public institutions (Table 1). Participants were selected by purposive sampling and met the following inclusion criteria: professionals working in public and social entities in the four above mentioned regions, whose professional tasks included assisting MSWs, and had a minimum of six months' experience working with them. The interviews lasted, on average, 60 minutes. Much of the information was collected face-to-face in different locations, although telematic interviews were also used due to the pandemic and the successive health restrictions.

### Data analysis

The interviews were recorded, and the information was transcribed verbatim for subsequent analysis with ATLAS.ti web. Five authors were involved in coding the transcriptions and conducting an inductive reflexive thematic analysis focusing on those aspects of the interviews relating to Covid-19 [50]. First, an initial code tree based on the interview script was designed and applied. After repeated reading of the interviews, the different units of meaning (sentences or paragraphs) referring to COVID-19 were identified and assigned the code "COVID-19". They were then re-read and inductively coded to create new emerging codes that described and summarized the meaning of the sentences or paragraphs. All resulting codes were grouped into code families according to their similarity, forming the first subcategories that allowed the identification of the final categories, subthemes and themes shown in the results. An example of the coding process could be consulted in Fig 1.

### Ethical considerations

We obtained the approval from the *Ethics Committee for Research with Medicines of the University Hospital Arnau de Vilanova- Lleida Territorial Management* (CEIC-2459). Interviews

**Table 1. Socio-demographic characteristics of the participants (n=92).**

| | |
|---|---|
| **Sex** | |
| Men | 37 (41%) |
| Women | 55 (59%) |
| **Professional profile** | |
| Health professionals | 17 (18%) |
| Social workers | 24 (26%) |
| Cultural Mediators | 8 (9%) |
| Members of NGOs | 18 (19%) |
| COVID-19 Trackers | 2(2%) |
| Trade unionist/worker representatives | 14 (15%) |
| Political officers | 8 (9%) |
| Businessmen | 2 (2%) |
| **Province** | |
| Almería | 16 (17%) |
| La Rioja | 20 (22%) |
| Lleida | 24 (26%) |
| Huelva | 11 (12%) |
| Jaén | 4 (4%) |
| Murcia | 17 (18%) |
| **Age** | |
| 18-25 | 1 (1%) |
| 26-40 | 30 (33%) |
| 41-50 | 36 (39%) |
| 51-65 | 25 (27%) |

were digitally recorded after obtaining verbal and written consent from the participants, who were informed of the study's aims and were guaranteed anonymity and confidentiality in expressing their opinions.

## Rigor

The research report process followed the Consolidated Criteria for Reporting and Publication of Qualitative Research (COREQ) [51]. Methodological rigor was ensured throughout the study following the quality criteria of Guba and Lincoln [52]. Credibility, transferability, and confirmability were used as quality criteria to guarantee the rigor of the research process. Credibility was enhanced by the triangulation of the interpretation of the analysis results among different researchers and by providing verbatim quotes in the results section to support their joint interpretation. Dependability was guaranteed as five authors were involved in collecting and codifying the data independently. Then, they made an independent proposal for the results, discussing and agreeing later on the most appropriate one. Confirmability was enhanced as all the authors have lengthy experience in qualitative research and migration and by the heterogeneity of the participant profiles (social services, NGOs, health services, activism), all with broad experience in working with migrant farmworkers. Transferability was achieved through the description of the context and the participant profiles.

## Results

We structured the results into three main themes: 1) COVID's extraordinary experience of cross-sectoral collaboration, 2) Compliance or non-compliance of MSW with the preventive

| Meaning unit / transcript excerpt | Selected quotation | Open codes | Family of codes | Category | Subtheme | Theme |
|---|---|---|---|---|---|---|
| *Every year, you can see them sleeping in doorways, in older people's homes, in huge arcades, and you can see them sleeping there. Not this year. Okay, so this year, this has changed; they have changed the chip. We have all changed it because they were not allowed to sleep in the street, so we put more places in the hostel, although every year there were places. There have always been plenty of places, and there has been no problem, we thought: this year, there is not going to be anyone sleeping in the street; there cannot be anyone sleeping in the street.* | *This year, they were not allowed to sleep in the street, so we put more places in the hostel (...) there cannot be anyone sleeping in the street* | MSW sleep in the street before COVID  It was not allowed to sleep in the street  The city council increased the places in the hostels and shelters  Improvements in accommodations for seasonal workers | COVID and accommodations | The priority was that MSW were dispersed and had accommodation | Convergence and Synergy in measures taken by regional and local authorities | COVID's extraordinary experience of cross-sectoral collaboration |

**Fig 1. Example of the coding process.**

and containment measures of COVID-19?, and 3) COVID exacerbated common health and healthcare problems of MWS. Their corresponding subthemes, categories and subcategories can be seen in Table 2. These themes explain how in order to mitigate the impact of the pandemic, both public authorities and NGOs implemented extraordinary measures. However, adequate adherence to primary prevention measures (which included maintaining a safe distance, surface and hand disinfection, and the use of protective masks and personal protective devices) and secondary prevention measures (screening, early diagnosis and detection of cases, and case and contact control aimed at breaking chains of transmission) was hampered by several factors. These obstacles included the inability to stop working, poor housing conditions (water and electricity shortages, housing location, and overcrowding), and challenges related to cultural and linguistic differences. These circumstances impacted the health of seasonal workers and their access to healthcare services.

## 1. COVID's extraordinary experience of cross-sectoral collaboration

In order to mitigate the impact of the pandemic in MFWs, local and regional authorities and NGOs implemented extraordinary measures, sometimes coordinately, which had rarely happened before.

**1.1. Convergence and Synergy in measures taken by regional and local authorities.** *The priority was that MSW were not dispersed and had accommodation.* Local and provincial public administrations required farmers to provide decent housing conditions for workers to comply with safety measures. Even so, given the inadequacy of the accommodation and the difficulty of converting it, the administrations set up temporary accommodation facilities for them, such as shelters. This measure was also aimed at preventing some of the MSW from finding themselves destitute and homeless.

*This year, they were not allowed to sleep in the street, so we put more places in the hostel (...) because we think there cannot be anyone sleeping in the street (Interview 30, City Council, La Rioja).*

Because of the overcrowding of these housing facilities, outbreaks occurred because MSW could not apply the recommended and established biosecurity recommendations on the need

**Table 2. Themes, subthemes, categories and subcategories.**

| THEME | SUBTHEME | CATEGORY | SUBCATEGORY |
|---|---|---|---|
| 1. COVID's extraordinary experience of cross-sectoral collaboration | 1.1. Convergence and Synergy in measures taken by regional and local authorities | 1.1.1. The priority was that MSW were not dispersed and had accommodation<br>1.1.2. Accommodation for isolation in case of positive or close contact | |
| | 1.2. Solidarity Initiatives taken by NGOs during the pandemic: prevention, food and hygiene | 1.2.1. Health education<br>1.2.2. Food and hygiene | |
| | 1.3. Health system efforts to screen and vaccinate MWS | 1.3.1. Screening strategies<br>1.3.2. Vaccination strategies | |
| | 1.4. Lessons learned and good practices of cross-sectoral collaboration and networking | | |
| 2. Compliance or non-compliance of MSW with the preventive and containment measures of COVID-19? | 2.1. Challenges in practice to implement the prevention measures | 2.1.1. Maintaining the safety distance between MSW | At work<br>In accommodation |
| | | 2.1.2. Hygiene, disinfection and MSW's use of protective face masks | At work<br>In settlements |
| | | 2.1.3. Performance of screening tests | PCR testing among MSW<br>Communication of PCR results to MSW<br>Registration of PCR results from MSW |
| | 2.2. Grounds for MSW non-compliance with the preventive measures | 2.2.1. MSW concealed symptoms for fear of losing work time and/or being evicted from the home | |
| | | 2.2.2. Cultural barriers and different conceptions of health/disease between MSW and the native population | |
| 3. COVID exacerbated common health and healthcare problems of MWS | 3.1. Imprints on physical and mental health of MSW | 3.1.1. Physical health<br>3.1.2. Mental health | |
| | 3.2. The Odyssey of Care for MSW: Access and quality of healthcare | 3.2.1. Neglect of other diseases | |
| | | 3.2.2.Telemedicine: increasing communication barriers and decreasing trust in the health system | Language barriers and diminished confidence<br>Increasing communication barriers |

to maintain safe interpersonal distance. It led to a reassessment of the maximum permissible capacity of the shelters, consequently reducing the number of infections in these facilities.

*For seasonal workers, shelters are opened that can accommodate many people. So if you make the hostel have a capacity for 20 people, if only one person is positive, they will all be positive and end up getting infected (Interview 8, Healthcare, Lleida).*

However, there was less government intervention in the permanent settlements. Even so, during the pandemic, resources such as drinking water taps, portable toilets, and rubbish collection were also installed. Participants recognized that most of these facilities were not maintained over time.

*/. . ./ with the money they received, it appears that what they were doing was collecting rubbish, but it's all over now. We are back to the initial situation (Interview 72, Huelva, NGO).*

*Accommodation for isolation in case of positive or close contact*

For those people who had to remain in isolation because they were positive or in close contact but did not have adequate housing, the different institutions provided houses, shelters, hostels, or prefabricated dwellings for their isolation.

*The hotel was also helpful for those who did not have a home where they could isolate themselves in conditions (. . .) because many lived in houses with many people. There were cases of seasonal workers who lived in a house with 15 other people (Interview 13, Healthcare, Lleida).*

However, the facilities did not always meet adequate conditions for this, nor were the seasonal workers properly informed of their situation, rights and obligations. Participants reported violent episodes with MSW during their stay in these dwellings. Also, there were cases of escape attempts because they did not understand why they had to stay there without being able to go to work, or because they did not agree to comply with the prescribed isolation or quarantine measures, and the state security forces had to intervene on some occasions.

*They were confined to sports centers or areas that were going well. Still, there were situations of risk and riots because they did not understand the sense of lockdown, a young person who emigrated from Senegal, who arrived here, and who is well despite being PCR positive (Interview3, Healthcare, Lleida).*

**1.2. Solidarity Initiatives taken by NGOs during the pandemic: Prevention, food and hygiene.** The solidarity initiatives taken by NGOs during the pandemic included health education about COVID to MSW and the provision of food and hygiene measures.

*Health education.* Since NGOs worked in settlements and with the seasonal population before the pandemic, they often facilitated bringing the health system closer to the most vulnerable populations. On their own initiative or in collaboration with agricultural companies or the health system, they educated seasonal workers on prevention and case-control measures and close contacts, as well as providing materials in different languages. To make health education more efficient among MSW, NGOs and administrations used professional mediators, people from different backgrounds, or recognized figures in the community to overcome language and cultural barriers.

*In the beginning, the problem we had with the seasonal workers was, on the one hand, the language issue, since we often did not understand each other (. . .) The mediators did a lot of work since language is essential to understanding each other because if you tell them to do isolation. They do not know what it is because they do not understand you and will not do it (Interview 8, Healthcare, Lleida).*

*Food and hygiene.* NGOs provided food to the settlements and set up soup kitchens for seasonal workers. They also facilitated disinfection by providing the necessary materials such as water, masks and disinfectants in the settlements to prevent further spread of the virus.

*It is tough to carry out disinfection in the settlements. So, we have done it in a micro way, training them, and then in some places, we have done something more industrial with one of these sulfate vats that we have rented from a farmer, we have poured the bleach tablets in, and we have passed it around (Interview 12, NGO, Huelva).*

*We have also distributed blankets, hydroalcoholic gels, masks /. . ./ We distribute everything and food to the settlements (Interview 25, NGO, Murcia).*

**1.3. Health system efforts to screen and vaccinate MWS.**   *Screening strategies*. In order to bring the prevention and control measures foreseen in the different action protocols closer to the population, the health system activated mobile units to the settlements and other areas frequented by MSW to carry out PCR diagnostic tests and distribute hygiene and protection materials to them.

*We set up a little stall where you could take a test if you wanted to; with all the guarantees, you could ask questions, make masks, material. . . (Interview 43, Healthcare, Murcia).*

However, health professionals in health centers sometimes lacked the time in their working day to go to the settlements or rural areas where MSW lived or worked, so in some communities, staffing was increased to care for the seasonal population.

*Now [during the campaign period], of course, the population doubled. So the problems doubled, so in the extensive health regions, they increased the number of social workers, which was not even considered in the years before the pandemic (Interview 13, Healthcare, Lleida).*

*Vaccination strategies*. Similarly, health institutions agreed on the importance of vaccinating the seasonal population and promoted outreach strategies. In some cases, vaccination was compulsory in order to be able to work, something that was not the case in other professions.

*They are not allowed to choose whether they want to be vaccinated or not /. . ./ The seasonal worker is a kind of cattle that comes, and we must vaccinate them so they do not bring diseases to the village (Interview 14, NGO, Lleida).*

Initially, seasonal workers were reluctant to be vaccinated, but this gradually changed as the pandemic progressed and more and more people in their environment became infected. Likewise, vaccination was particularly difficult for seasonal workers in an irregular administrative situation. Hence, placing vaccination points close to their place of work or residence was once again influential in getting them vaccinated.

*In the beginning, many people were reluctant to get vaccinated. When they saw that there were positive cases, the number increased. When I went to vaccinate directly, people who first said no signed up (Interview 80, NGO, Huelva).*

**1.4. Lessons learned and good practices of cross-sectoral collaboration and networking.**   Participants working in public institutions, having planned and implemented real-time measures for an unexpected pandemic, described that learning and continuous reviewing protocols would improve performance in subsequent campaigns.

*[The next campaign] will be COVID, but fear will no longer exist. We will be working with protocols that we have already studied and evaluated from the previous year, so I hope it will be an even simpler campaign (Interview 30, City Council, La Rioja).*

The participants considered the mobilization and joint work between public and social institutions to be exceptionally positive, considering it critical to the correct application and continuous revision of the protocols. They also pointed out the importance of companies getting involved in applying measures to prevent contagion among workers, as this significantly reduced the number of contagions.

*We have been working in an intersectoral way with many administrations. We have been working, as well as being with the population, with the farmers, with the technicians, the health technicians, myself and another nurse, the social services, and the health centers. . . There were also the local administrations, Town Councils from all areas, the Local Police, and the Civil Guard. . . We were all there, in the meetings! (Interview 33, Healthcare, La Rioja).*

Likewise, the increase in inspections by the administrations prevented overcrowding and improved accommodation conditions. Moreover, increasing the number of places available in shelters has prevented people from sleeping rough, as has been the case in previous years.

*Now, due to COVID, there is a prior review of accommodation. Because before, the farmer would say, "I have accommodation," and that was it (Interview 38, City Council, La Rioja).*

## 2. Compliance or non-compliance of MSW with the preventive and containment measures of COVID-19?

About seasonal workers' compliance with the rules established for the containment of Covid-19, participants shared two types of discourses: one that identifies them as good at complying with the rules and the other that attributes incorrect compliance with the measures to them.

**2.1. Challenges in practice to implement prevention measures.** *2.1.1. Maintaining the safety distance between MSW*. **At work.** Informants explained that companies established guidelines so that, even in the open air, seasonal workers were kept at a certain distance at all times. In addition, work and transport to the fields were planned based on bubble groups, in which they worked with individual teams.

*We tried to take the measures that we thought best suited us: making bubble groups, people who didn't work with each other, we divided them according to the vans. . . Each van went to work in one place and then they slept together. They didn't mix with the others, working different schedules, temperature controls. . . Look, then we did tests on them, we did tests on ourselves, we were constantly doing them, we helped them to do the shopping so that they didn't have to go to the supermarket, which was considered to be an important risk factor. . . (Interview 62, entrepreneur, La Rioja).*

Meanwhile, other companies did not set guidelines to respect social distance in the workplace, even in closed environments.

*In September 2020, I visited farms where workers did not keep their social distance and the companies did not provide them with masks or hydroalcoholic gel (Interview 77, trade union, Murcia).*

However, the continuous movement of workers between localities and autonomous communities facilitated the transmission of the virus, reducing the effectiveness of the measures taken.

*In accommodation*. Due to the substandard and overcrowded conditions in which the most vulnerable MSW live, it was difficult, if not impossible, for them to respect the safety distance between people and quarantine measures recommended during the pandemic. However, the continuous movement of workers between localities and autonomous communities facilitated the transmission of the virus, reducing the effectiveness of the measures taken.

*Isolation and everything are perfect, but that's for those who can afford it. Where will they go if they don't even have a hut? Are they going to go to a place where 10 people live? Where*

*there is a bathroom for each of them. . . It makes me laugh (Interview 49, Healthcare, Almería).*

*2.1.2. Hygiene, disinfection and MSW's use of protective face masks.* **At work.** Just as labor regulations stipulate that the contracting companies must provide safety devices at work, the companies did not provide them with masks or hand disinfectant products in several cases.

*And they came to us, above all, with complaints like: "hey, in the companies they don't give me this, in the company they don't give me that" (Interview 41, trade union, Murcia).*

Even when they did receive them, they were sometimes dirty or in a condition that made them practically unusable. In addition, the workers did not have the money to buy masks either, not least because of the high prices and low accessibility of the masks.

*/. . ./ some people are working less than a meter away, some wear masks, and others wear a handkerchief because they can't afford masks (. . .) You can see from a distance that it is dirty, but they want to buy masks for their children who go to school and prefer to reuse them several times. But it's expensive to maintain safety, and they have to take it on themselves (Interview 40, NGO, Murcia).*

**In settlements.**   For the most vulnerable MSW living in farmhouses or shacks, measures such as hand washing, and surface disinfection were practically impossible to carry out due to the lack of running water.

*2.1.3. Performance of screening tests.* **PCR testing among MSW.** Health institutions encountered resistance or desertion to take the PCR test, as testing positive meant risking the possibility of working or being under the control of the administration. Furthermore, traveling to take the PCR test was another reason why people chose not to take the test, even when transport was free. Sometimes, people who refused to take the test were also ineligible for other services provided by the associations, such as shelter or canteen, or for employment through the administration or trade unions, as taking the PCR was a prerequisite.

*"You don't do PCR on me". So nothing. Well, just so you know, then there is no work, because the objective was to ensure that the COVID issue would not spread any further. (Interview 31, trade union, La Rioja).*

Some MSW considered it essential to know their status and adapted well to measures to protect their health. If they were free of COVID-19, it was used as a guarantee of being able to work, even if they also did so with a particular fear of being positive.

*We said: "There will be people who don't want to take the test" . . . on the contrary! they were willing to take the test. It was a guarantee for them. Do you understand? So they have adapted perfectly. (Interview 30, City Council, La Rioja).*

**Communication of PCR results to MSW.**   Since PCR results were not obtained on the spot, there was great difficulty in communicating the results when they were positive due to the high mobility of the seasonal population, the frequent number of telephone changes, the concealment of the MSW themselves so that they could not be located and prevented from working, or the use of the same documentation by more than one person.

*They go to work and get tested, and you have to give them the result the next day. If it takes them two days to leave, they may be no longer at the shelter, and by the time you give them the result, they will have already left. OK, you also have a lot of difficulties with mobile phones to locate them. And many challenges with documentation that I did not imagine: They bring you a photocopy of the documents (. . .) it is not usual, but some people have used the documentation of others, that is, several people are using the same documentation (Interview 68, social services, La Rioja).*

**Registration of PCR results from MSW.** Registration in the health system program was also administratively complex, as the program was not prepared to take records from people without a health card or those who were not previously registered in any national health system. Also, name registration errors resulted in duplicate registration of the same person.

*In the end they are very long names, with a lot of vowels and consonants that you don't register a mistake and it can be another new person and you can open two health histories for one person (Interview 68, social services, La Rioja).*

**2.2. Grounds for MSW non-compliance with the preventive measures.** *2.2.1. MSW concealed symptoms for fear of losing work time and/or being evicted from the home.* The administration's measures stated that the person should not go to work after testing positive, being in close contact with a positive case, or having symptoms compatible with COVID-19. Therefore, some people concealed their status in order to be allowed to continue going to work.

*[They] say: We don't tell because then they confine me for 14 days or 10 days. I lose my job if I say it to the boss, and he can fire me, and if I don't go to work, I can't eat. So, well, I get by as best I can. I don't say anything if my health allows me to, and that's it (Interview 49, healthcare, Almería).*

Thus, participants identified that MSW experienced the COVID phenomenon as an impediment that would not allow them to work. In hiding these conditions, they deployed different strategies to deny being screened or identified as positive by the administrations, such as hiding or changing their telephone number to avoid losing their job opportunities.

*[They say]: "And if I test positive, I cut the phone, and they can't find me. And when they go to where I used to live, the others say I don't live there anymore. Or if you call, I say no, that I'm fine, that I don't need it and don't interfere in my life. . .. "(Interview 49, healthcare, Almería).*

For all these reasons, the most significant identification difficulties were with the temporary population in an irregular administrative situation, without a contract, and those living in settlements. They needed more resources, were more afraid of being identified, were more challenging to locate, and depended more on work for their livelihoods.

*This virus and everything in life distinguishes us by class, and it always affects the most vulnerable people more because of the housing conditions, the hygiene conditions, the working conditions. . . (Interview 11, NGO, Lleida).*

In households, MSW developed defensive strategies such as expelling infected people from the house or work crew, denying that they lived there to protect them, and avoiding having to be quarantined as close contacts.

*When we have detected one, it is that. . . they are kicked out of the house by their own companions. Because, of course, as they say, "Oh my God, this guy, he's going to infect us all." /. . ./ They also keep quiet because they say, "Oh my God, if my colleagues find out, I'll lose my housing, they won't hire me. . ." (Interview 2, healthcare, Almeria).*

*2.2.2. Cultural barriers and different conceptions of health/disease between MSW and the native population.* The participants found that the fact that the virus did not affect them in principle in their daily lives made COVID seem unimportant to them. Moreover, due to misinformation about the mode of infection and the severity of COVID-19, seasonal workers denied the possibility that COVID-19 could affect them, or at least not significantly.

*So the COVID thing sounds like Greek to them, I mean, is that going to get me? What are you saying? What is that, a toothache? (Interview 27, NGO, La Rioja).*

*When you tell him: "I have bad news for you, you have COVID", do you know what it means to him: "What do I have COVID? /. . ./ "I would be dead and that is not possible. This is a lie" (Interview 32, NGO, Lleida).*

Similarly, people who had experienced hardship in their migration journey and those who had lived with various serious endemic diseases in their countries did not attach importance to a disease with (a priori) mild symptoms.

*This is where a cultural factor comes in. And the whole issue of COVID when they have had such a risky and life-threatening migratory journey, arriving here and being confined and locked up for two weeks. . . they don't understand it (Interview 3, Healthcare, Lleida).*

Therefore, in asymptomatic or mildly symptomatic positive cases, they did not see the need to follow the guidelines set by the administrations, even less so if they implied a mobility limitation and consequently had to stop working. However, in some cases, the perception of the risk involved in COVID changed as they saw cases of people close to them who were severely affected as a result of infection.

*Throughout the confinement, they said: "I thought that my race was not vulnerable to the virus." But when they have seen that two or three of their companions are admitted to the ICU with bilateral pneumonia, then it is clear that their race is also vulnerable. (Interview 80, NGO, Huelva).*

## 3. COVID exacerbated common health and healthcare problems of MWS

The health problems related to the pandemic, added to the health problems already faced by MFW due to their social and labor conditions. Moreover, access to healthcare services was complicated due to communications problems.

**3.1. Imprints on physical and mental health of MSW.** *3.1.1. Physical health.* The informants explained that the seasonal population has health problems that they carry with them due to their living and working conditions, which are marked by vulnerability. They also pointed out that the incidence of COVID among the seasonal population, in many cases, was not as high as the administrations and associations might have expected at first.

*The impact of COVID is one more problem added to everything they already have regarding health. But the incidence has been less than we thought as well (Interview 48, NGO, Almería).*

*3.1.2. Mental health.* Due to the lack of support networks in Spain and the fact that in their countries, the pandemic was also affecting their families and relatives, the mental health of MSW also suffered. In addition, people placed in isolation centers for having been positive or in close contact with a positive person also felt sadness and loneliness because they were unable to communicate with their families due to a lack of digital devices.

> *I have met many men who burst into tears in the consulting room "because my father died tonight". Many deaths of family members in Morocco who have had to live in the loneliness of not being able to go to see him (Interview 17, NGO, Almeria).*

**3.2. The Odyssey of Care for MSW: Access and quality of healthcare.** *3.2.1 Neglect of other diseases*. Due to COVID, there were hardly any face-to-face consultations in health centers, which meant that the health protocols for specific attention to immigrants, through which serious illnesses can be detected even in a latent or asymptomatic state, were not applied.

> *It is a pity because primary care doctors, for example, used to ask for the immigrant care protocol, where they detected a lot of asymptomatic hepatitis; they caught HIV early without giving rise to the development of AIDS. . .. And that is now being lost (Interview 4, healthcare, Almería).*

3.2.2. *Telemedicine*: *increasing communication barriers and decreasing trust in the health system*. **Language barriers and diminished confidence.** The language was a significant barrier to communication for the administrations to carry out health education or tracing work among MSW, even more so when communication was by telephone. The fact that health care was provided by telephone rather than face-to-face during the pandemic decreased the accessibility of health services for MSW. In addition, it was less efficient in diagnosing or following up on different ailments due to communication and language barriers. All this led to a decrease in the confidence of seasonal workers in the health system, as they felt that it did not answer their health problems.

> *On top of that, appointments are made by telephone. Of course, this causes a lot of mistrust. If my doctor calls me on the phone, he won't understand what I have, and I want him to see me and be able to express it. Well, a person who does not speak the language well and is unable to express himself has been quite difficult (. . .) People stop trusting the system, right? If the doctor is not going to call me, he is not going to see me and will not solve anything (Interview 6, NGO, Almería).*

## Increasing communication barriers

f the digital divide among immigrants already existed before COVID-19, the fact that they did not have a digital device from which they could make an appointment for health care or receive a call made accessibility to health consultations even more complicated.

> *The digital divide is enormous. In other words, we take it for granted that everyone has a smartphone, a computer, and wifi, but nothing could be further from the truth. To ask for an appointment like I do, using a mobile phone or a laptop. That is unthinkable. This is also a brutal barrier, already a barrier before and even more so in COVID (Interview 60, NGO, Murcia).*

Despite the difficulties, NGOs acted as "interpreters" in answering health calls. Therefore, the NGOs pointed out the vital importance of the existence of cultural mediators in health centers and their involvement in both face-to-face and telephone consultations.

*I have had to call the doctor and ask her: look, this case of this woman has not understood anything of what you have just told her, and she has had to explain it to me so that I can explain it to her /. . ./ with the issue of COVID, for me it would be essential to have intercultural mediators, especially in health centers (Interview 46, NGO, Almería).*

## Discussion

This study has highlighted, from the point of view of health and social professionals, the difficulties faced by MSW in complying with prevention measures during the COVID-19 pandemic and the impact this had on their overall health. Among the various problems identified were the precarious situation of MSW, high mobility, displacement, overcrowding, fear of losing their jobs, stigma, and language and cultural barriers. These difficulties are related to multiple social asymmetries linked to living and working conditions and migration structures, processes, and policies, structural elements that can be interpreted as the first obstacle that hindered the measures' implementation.

In the context of these structural inequalities, the agricultural sector, largely dependent on the migrant population, continued to need labor during the pandemic, directly affecting the effectiveness of the measures adopted by the authorities. However, the central government, regional governments, trade unions, and third-sector organizations were forced to implement containment measures to manage demand in a pandemic scenario. These sought to facilitate the mobility of workers within the country and the arrival of regular migrant workers from Eastern Europe and North Africa [16]. However, despite efforts to meet the sector's needs, some employers opted for irregular recruitment of MSW, mainly from the Maghreb and sub-Saharan countries [53]. As the results of this study show, this dynamic created a complex scenario in a pandemic context, as the combination of the urgency of the demand for labor, mobility restrictions, and the irregular hiring of workers affected the effective implementation of preventive measures and ended up affecting the physical and mental health of workers. Moreover, the arrival of both regular and irregular workers posed additional challenges in implementing and monitoring preventive measures.

Thus, the COVID-19 crisis has exposed, from a European perspective, the deep-rooted structural problem in the continent's agro-industry, especially in the south, where migrant labor operates under precarious conditions [45]. In 2020, the European Commission issued guidelines to facilitate the mobility of workers deemed essential [54]. Additionally, the New Pact on Migration and Asylum addressed migration as a key issue for the EU and offered a common framework for managing migration and asylum based on solidarity among all member states [55]. These initiatives focused more on managing irregular migration and prioritizing market demands than on improving rights and safeguarding the health of migrants, specifically MFWs [46]. In countries like Italy, actions were limited to the regularization of irregular migrant workers and the implementation of a "health pass" as a requirement for employment [56]. This resulted in new roles and responsibilities for migrant workers, paving the way for the regularization of their status [57], but without substantial improvements in their social and working conditions in the short or medium term [56]. On the other hand, Germany and Austria opted to allow the employment of migrants in agriculture and care, exempting them from travel bans, measures criticized for not ensuring adequate health protections

for workers and being more oriented towards ensuring food availability for the entire population [58, 59]. In the UK, following the lockdown, agricultural organizations voiced concerns about the economic impact on agriculture due to the lack of labor, and the solution involved facilitating the importation of labor from countries like Romania and Bulgaria [60].

Shifting our attention back to the findings of this research centered around the Spanish example, one of the patterns identified in the measures implemented was job insecurity and precariousness of life linked to the irregular administrative situation. This condition led to a shadow economy characterized by substandard wages and abusive business practices, among others, which fostered inequality and inequity between native citizens and immigrants [61]. This group's need to remain active to secure income led workers to hide some of their symptoms for fear of losing employment opportunities. As a result, they were likely to remain in informality and consequently out of reach of health and care services [62]. According to informants, the absence of access to health and care services exposes these irregular migrant workers to more significant difficulties in addressing and treating their health problems, potentially contributing to the spread of disease and the aggravation of pre-existing medical conditions. In the context of the COVID-19 pandemic, the lack of adequate health care impacted individuals and had broader public health implications by making these MSW potential disease vectors.

In this context, mass vaccination, as one of the significant pandemic containment measures, was ineffective in high-incidence and mobility scenarios. Personal protection measures were the best allies in breaking the chains of transmission and thus reducing the incidence of cases [63]. As the results show, the workers sometimes had to procure measures and personal protective equipment to protect themselves and their relatives or household members from suspected or confirmed cases [64]. Employers' ineffective distribution of protective materials was a challenge for professionals in the sector and, by extension, for the population as a whole [65, 66].

Regarding health care, the fear of detention or deportation emerges as an element that influences the reluctance of migrant farm workers in irregular situations to seek health care when they need it. Not only does this fact contribute to the dynamics of exclusion, but the continuous accumulation of psychological stress can trigger an allostatic load, negatively affecting general health and increasing vulnerability to chronic diseases over time [67–69].

The situation of MSW in Spain, especially during the COVID-19 pandemic, has highlighted the vulnerabilities and challenges faced by these essential workers in the agricultural sector. Before the pandemic, MSW already confronted precarious working conditions, including limited access to healthcare, inadequate housing, and difficulties navigating legal and social services, often exacerbated by language barriers and uncertain immigration status [70]. The COVID-19 pandemic intensified these inequalities, further complicating compliance with preventive measures and exposing MSW to more significant health risks. Local authorities and NGOs implemented extraordinary measures to mitigate the impact, such as temporary accommodation, mass testing, and stricter labor regulations. Still, working conditions, constant mobility, precarious housing, and cultural and linguistic barriers remained significant obstacles to effective preventive measures [71].

The political and administrative response to the situation of MSW during the pandemic has been a subject of debate. In 2018, Spain adopted a national law that significantly expanded access to healthcare for all residents, including undocumented migrants, marking a substantial shift from a more restrictive coverage system in previous years [72]. However, irregular migrants continued to face challenges in accessing healthcare, underscoring the need to address legislative and administrative barriers for the effective implementation of this law. Additionally, the pandemic has highlighted the agricultural sector's dependence on MSW and the importance of ensuring their rights and well-being, not only as a matter of social justice but also for the country's food security [71].

Finally, the ignorance of authorities, professionals, and NGOs about the needs and complexities of this group was an element that hurt decision-making and the implementation of case and contact management measures. For example, it was reflected in the lack of follow-up programs and insufficient intercultural and structural training of health personnel to care for a diverse migrant population with problems that went beyond the pandemic [73]. Public health measures failed to reach the migrant community effectively due to several barriers. These included the lack of dissemination in several languages, the implementation of telemedicine in a context with language, cultural, and digital barriers, the lack of knowledge of the daily practices and issues of the community, as well as difficulties in access and two-way communication between public health and MSW. These limitations could have deepened the mistrust towards protocols, devices, and social and health workers, which needs to be addressed in future research.

## Implications

This study highlighted the imperative need to foster and maintain over time collaboration between different entities, including administrations, social and health professionals, NGOs, and companies, to enhance the quality of care offered to MWS. The synergy between different actors is essential in effectively addressing beneficiaries' needs and challenges, thus ensuring a more comprehensive and efficient response.

This scenario underscores the importance of comprehensively addressing health disparities, providing valuable learnings and lessons in a pandemic, and caring for MSW during future agricultural seasons. In addition, the need to address health inequalities from a holistic perspective, recognizing the interconnectedness of social, economic, and cultural factors, has been highlighted. In this regard, the figure of the cultural mediator emerged as a crucial actor in facilitating communication and understanding between workers and public health measures.

## Limitations

Finally, it is critical to recognize the inherent limitations. As the study relies primarily on the narratives of professionals; there is a need to broaden the perspective by incorporating the direct voices of migrant farm workers themselves. It would allow a better understanding of their experiences, challenges, and needs. It also highlights the importance of moving towards continuous and participatory research involving workers, practitioners, and specific contexts. Such an approach would contribute to a deeper understanding of workers' health and shed light on underlying health disparities, thus promoting more informed and equitable interventions.

## Conclusions

The study underlines the close relationship between health and care for MSW of migrant origin and the social asymmetries that generate inequality. The pandemic has played an essential role in shedding light on the contribution of MSW and their work. It also revealed their essential contribution and the systemic vulnerability that characterizes their situation, a pre-existing but previously ignored reality. Lessons learned manifest in improved coordination between companies, social services, administration, and NGOs in caring for workers and increased inspections to ensure compliance with laws and trade union agreements. These lessons are not only relevant in the context of a pandemic but also reveal the need for deeper structural transformations to ensure that food is produced sustainably.

## Acknowledgments

The authors thank the participants in this study who kindly shared their experiences.

## Author Contributions

**Conceptualization:** Agustín González-Rodríguez, Sergio Andrés-Cabello, Erica Briones-Vozmediano.

**Formal analysis:** Iratxe Perez-Urdiales, María del Mar Jiménez-Lasserrotte, Mar Pastor-Bravo, Erica Briones-Vozmediano.

**Funding acquisition:** Agustín González-Rodríguez, Erica Briones-Vozmediano.

**Writing – original draft:** Miquel Úbeda, Vanesa Villa-Cordero, Iratxe Perez-Urdiales, Erica Briones-Vozmediano.

**Writing – review & editing:** Miquel Úbeda, Vanesa Villa-Cordero, Agustín González-Rodríguez, Sergio Andrés-Cabello, Iratxe Perez-Urdiales, María del Mar Jiménez-Lasserrotte, Mar Pastor-Bravo, Erica Briones-Vozmediano.

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
