## [Decision Letter · Decision Letter 0]

19 Feb 2024

PONE-D-24-02040Lessons learned from institutional responses to protect Migrant Farmworkers during COVID-19 in SpainPLOS ONE

Dear Dr. Perez-Urdiales,

Thank you for submitting your manuscript to PLOS ONE. After careful consideration, we feel that it has merit but does not fully meet PLOS ONE’s publication criteria as it currently stands. Therefore, we invite you to submit a revised version of the manuscript that addresses the points raised during the review process.

We look forward to receiving your revised manuscript.

Kind regards,

Cesar Infante Xibille, Ph.D

Academic Editor

PLOS ONE

Journal Requirements:

"This work was supported by the Instituto de Salud Carlos III (ISCIII), under the grant: PI20/01310, and co-funded by the European Union"

Please state what role the funders took in the study.  If the funders had no role, please state: ""The funders had no role in study design, data collection and analysis, decision to publish, or preparation of the manuscript."" If this statement is not correct you must amend it as needed. 

"Acknowledgments

The authors thank the participants in this study who kindly shared their

experiences and to the organizations that have funded the project"

Please be informed that funding information should not appear in the Acknowledgments section or other areas of your manuscript. We will only publish funding information present in the Funding Statement section of the online submission form. 

"This work was supported by the Instituto de Salud Carlos III (ISCIII), under the grant: PI20/01310, and co-funded by the European Union"

5. We note that you have indicated that there are restrictions to data sharing for this study. For studies involving human research participant data or other sensitive data, we encourage authors to share de-identified or anonymized data. However, when data cannot be publicly shared for ethical reasons, we allow authors to make their data sets available upon request. For information on unacceptable data access restrictions, please see http://journals.plos.org/plosone/s/data-availability#loc-unacceptable-data-access-restrictions. 

Reviewers' comments:

Reviewer's Responses to Questions

**Comments to the Author**

1. Is the manuscript technically sound, and do the data support the conclusions?

Reviewer #1: Yes

Reviewer #2: Yes

Reviewer #3: Yes

Reviewer #4: Partly

Reviewer #5: Partly

2. Has the statistical analysis been performed appropriately and rigorously? 

Reviewer #1: N/A

Reviewer #2: N/A

Reviewer #3: N/A

Reviewer #4: N/A

Reviewer #5: N/A

3. Have the authors made all data underlying the findings in their manuscript fully available?

Reviewer #1: Yes

Reviewer #2: Yes

Reviewer #3: Yes

Reviewer #4: No

Reviewer #5: No

4. Is the manuscript presented in an intelligible fashion and written in standard English?

Reviewer #1: Yes

Reviewer #2: Yes

Reviewer #3: Yes

Reviewer #4: Yes

Reviewer #5: Yes

5. Review Comments to the Author

Reviewer #1: This is an interesting study focused on exploring experiences shared by social and health professionals during the COVID-19 pandemic in four specific regions of Spain: Catalonia, La Rioja, Murcia and Andalusia. Summary: adequate, the methodological design needs to be introduced. Title: informative, does the term institutional responses encompass organizations such as NGOs? Introduction: broad and well presented. Authors describe the study problem well. Well justified and contextualized study. Methods: a qualitative descriptive study is an appropriate design for this work. Justify why the study was carried out in those 4 regions and not in others, would the conditions have changed? This may be a limitation of the study. What were the sample selection criteria? Although grouped sociodemographic data are given, inclusion/exclusion criteria are not clear. Define members of the sample well. Who are Members of associations/activists? Who are social integrators? Participation of all these groups in the sample should be better justified by explaining their functions in reference to these migrant workers. Is it purposive or convenience sampling? Braum & Clark's thematic analysis phase could have been used as it was a qualitative/descriptive design. Incorporate an example of the coding process. Authors present approval from the Ethics Committee for Research with Medicines of the University Hospital Arnau de Vilanova-Lleida Territorial Management (CEIC-157 2459); This is a local committee, does this authorize them to obtain data in other regions of Spain? Who designed the interview script? Who made the transcriptions of the interviews? The study does not make clear whether data saturation was reached. Well-developed rigor section. Use of COREQ checklist is important in these studies. Results: interesting, very explanatory, they broadly reflect the response to the stated objectives. The names of the categories are not very risky, it seems that the analysis was deductive. Discussion: I miss greater confrontation with results from similar studies in other European regions highlighting similarities/differences. Limitations of this study should be noted after the discussion. Conclusions. Although they respond to the objectives, authors should focus more on the specific conclusions of their study. The rest of the sentences should be included later in a section on lessons learned or implications for practice.

Reviewer #2: Dear editor

Thank you for the opportunity to review this article titled xxx

This is an interesting qualitative study on the perspective of professionals on the challenges posed by caring for immigrants who work in the Spanish agricultural sector.

The introduction is good. Define the topic appropriately, provide background and justify the study

Maatearials and Methods:

p. 5, line 129. The phrase "Wevconducted 87 semi-structured personal and online interviews with 93 participants" is confusing. If 87 interviews were carried out, how did the remaining 6 participate??? Or were there 93 interviews? Please clarify this and remove participants who did not participate in the interviews

Get the name of the analysis software correctly: it is ATLAS.ti (capital letters)

Line 153-154 (COREQ) should not focus on ethical issues but on rigor.

I suggest describing the data analysis based on the procedure described previously by one author. I suggest Braun and Clarck's reflective thematic analysis

See for example: Clarke, V., & Braun, V. (2017). Thematic analysis. The journal of positive psychology, 12(3), 297-298.

Results:

Please, provide a definition of the main categories before getting into the subtopic.

The explanations of some subtopics can be improved. For example: Food and hygiene has a very brief explanation and development.

The structure of Categories and subcategories is not clear. Some topics have two hierarchical levels and others have three. I suggest unifying this

An initial table or concept map from ATLAS.ti would help to obtain an overview of the themes and subthemes developed

Conclusions

I think the conclusions are too extensive and do not always derive from the results or respond to the objective. Perhaps some of them could be included in a subsection of implications for practice and policy.

Reviewer #3: The article focuses on the difficulties faced by migrant agricultural workers in complying with prevention measures during the COVID-

19 pandemic and the health consequences in Spain. The research carried out, using qualitative methods and conducted 87 semi-structured personal and online interviews with 93 participants interviews with involved professionals in social and health care: NGO 132 staff, health workers, social workers and educators, intercultural mediators, agricultural trade unions, and people with positions in public institutions. Particular emphasis is given on the consequences, their working conditions and perceptions. This is a very pertinent issue and the article deals well with this.

General

The fit between, theory and empirical analysis is well developed, and aids the author(s) in his/her/their analysis, and in framing the conclusion. The main literature mentioned covers the range of the research. The manuscript is informative regarding this particular issue in Spain, while contributing in relevant international research and literature.

The author(s) addresses a significant research subject and presents interesting field material. This article could be a starting point for further research.

Introduction

A clear stating and focusing of the argument is provided. It offers a robust theoretical frame.

Methods

Methods are appropriate and the fit between theoretical discussion and methodology is well formulated.

Results

Results are linked suitably to the other sections of the article. A well-organized and compelling discussion of the results is provided as well. The results are of interest for practice, social and migration policy and society more generally.

Conclusions

The conclusions are linked to the hypothesis and background characteristics incorporated into the results.

Language

The manuscript is informative and its reading enjoyable. The author(s) has/have paid attention to the clarity of expression and readability, such as sentence structure.

Reviewer #4: This paper examines the response of support staff, NGOs, service providers and various state actors to migrant agricultural workers in Spain during the COVID 19 pandemic. The paper employs interviews with the above participants to address the initiatives undertaken to address migrant workers health, to minimize their risk and exposure to COVID19 and the challenges in implementing these initiatives. The paper argues that the structure of the program, “irregular administrative situation” (line 599), ignorance of support workers and the recruitment model inhibited support for migrant workers during this time.

The paper provides a valuable framing of this issue as there is need to document the support landscape for MAWs both prior to, during and post Covid19. It provides some evidence to support its findings by including quotes from various staff and support workers in healthcare. The paper is somewhat descriptive, but its contribution to scholarly and practical knowledge of the topic would benefit by providing more content and to a lesser degree to situate the study in current knowledge of the impact of COVID 19 on this workforce. The former is particularly important to ensure its conclusions and recommendations are convincing and useful.

The paper would make a greater scholarly contribution by more clearly situating the experience of MAWs in migration studies or health debates, either in the introduction or conclusion, an orientation that would make the results more transferable. The precarious nature of MAWs as the paper describes, is already well documented. However, why is it important for us to know about how MAWs were supported during COVID19? What was the existing support landscape for MAW in Spain? There is less known about the latter topic and providing more clear structural description of support would greatly enhance how scholars in this field can make sense of support for temporary agricultural workers.

The paper could provide a more detailed discussion of the policy and administrative context for migrant labour in Spain, including the rights and benefits they have to healthcare prior to the pandemic and working conditions for workers (for example, how workers are contracted and what they do). Also, how participants vary in their pre-existing knowledge of, and contact with MAWs. This is important given that the paper concludes that the MAW program itself inhibits workers access to support (lines 474, 588), and that ignorance of MAWs was a factor in inhibiting their support. What for example is meant by “irregular administrative situation?”. What was the existing support/service landscape for MAWs prior to COVID 19? What were the responsibilities of employer/growers and was this laid out in work-contracts?

The paper also provides virtually no information about the MAW population in Spain. How many workers are there on average? Where do they come from? We only learn they are from Eastern Europe and North Africa on line 588 at the end. While the paper does acknowledge its limitation in not having interviewed MAWs, this does not preclude describing this population. This is important given that ‘cultural’ issues and language are raised as barriers and identified as important areas for increased support. The paper interprets ‘cultural issues’ described by a participant into ‘hardship in their migration journeys’, (lines 485-488). The description here is however very thin. Moreover, the paper suggests that MAWs responses to COVID initiatives were informed by their prior experiences but does not really discuss who workers are and what these experiences might be.

Some description of the conditions in which MAWs work and live as well as the policy context is important here because it highlights the challenging context in which service and support can be implemented in future.

Reviewer #5: Thank you for the opportunity to review this article. On the one hand, this is an important and relevant topic well justified by the authors. On the other hand, some areas need to be explained to a global audience, and a more in-depth explanation would be appreciated.

Although the focus is the migrant seasonal workers' vulnerabilities during the peak of the COVID-19 pandemic, the authors choose to interview and work with other populations as NGOs, and there is no work with the MSW. It is not clear why they made this decision it seems as if others talking about MSW experiences, so why not include them?

Who are these migrant seasonal workers, from where, and how does the seasonal work imply?

How the authors decided where to do the interviews (Comunidad), where does the interviews where done? How did they decide to include those groups of participants and no other ones?

In some parts of the results section, the discourses are about seasonal workers. It is not clear if the quotes are referring to migrants or other seasonal workers.

The analysis section treats migrants as if there is only one kind of meigration when there are very different SDHs for each migratory context, and there is a lack of information about different politics in each region in where the interviews were done. COVID-19 regulations and migration flows are different for each of them, and it is not reflected in the text.

There is an important number of subcategories this make difficult to center on the focus of the study, the main goal is diffuse.

Codes for the quote’s interviews are not clear. For example, what does the “e” mean?

In the conclusions section l. 657-659 the authors explain how migration policies need to be re-evaluated, but it is not clear which ones (and in where) they are referring to as there a lot of different issues along the text.

6. PLOS authors have the option to publish the peer review history of their article (what does this mean?). If published, this will include your full peer review and any attached files.

Reviewer #1: No

Reviewer #2: No

Reviewer #3: No

Reviewer #4: No

Reviewer #5: No

---

## [Author Response · Author response to Decision Letter 0]

20 May 2024

Response letter

PONE-D-24-02040

Lessons learned from institutional responses to protect Migrant Farmworkers during COVID-19 in Spain

Dear Editor of PlosOne,

We appreciate the comments given by the editor and the reviewers, which we think sharpens the manuscript on several important points. Please, find below an explanation of how we have addressed the comments in the response letter. Changes in the manuscript have been marked using the track changes function. 

Kind regards,

Iratxe Pérez-Urdiales

Corresponding author

- Format of the comments:

Reviewers’ comment in bold

Author comments in normal typeface

 Excerpts from manuscript italicized and indented

The format of the text has been revised to comply all the requirements included in the templates

Thanks for noticing. Funding information has been changed in all the sections

"This work was supported by the Instituto de Salud Carlos III (ISCIII), under the grant: PI20/01310, and co-funded by the European Union"

Please state what role the funders took in the study. If the funders had no role, please state: ""The funders had no role in study design, data collection and analysis, decision to publish, or preparation of the manuscript."" If this statement is not correct you must amend it as needed. 

The statement has been included in the cover letter and the cover letter has been resubmitted

"Acknowledgments

The authors thank the participants in this study who kindly shared their experiences and to the organizations that have funded the project"

Please be informed that funding information should not appear in the Acknowledgments section or other areas of your manuscript. We will only publish funding information present in the Funding Statement section of the online submission form. 

"This work was supported by the Instituto de Salud Carlos III (ISCIII), under the grant: PI20/01310, and co-funded by the European Union"

No funding information appears in the Acknowledgment section. And funding information has been amended in the platform and including a new statement in the cover letter. 

5. We note that you have indicated that there are restrictions to data sharing for this study. For studies involving human research participant data or other sensitive data, we encourage authors to share de-identified or anonymized data. However, when data cannot be publicly shared for ethical reasons, we allow authors to make their data sets available upon request. For information on unacceptable data access restrictions, please see http://journals.plos.org/plosone/s/data-availability#loc-unacceptable-data-access-restrictions. 

Reviewers' comments:

5. Review Comments to the Author

Reviewer #1: This is an interesting study focused on exploring experiences shared by social and health professionals during the COVID-19 pandemic in four specific regions of Spain: Catalonia, La Rioja, Murcia and Andalusia. 

Summary: adequate, the methodological design needs to be introduced. 

Thanks for the appreciation. The methodological design has been included in the abstract: (line 36-44)

This descriptive qualitative study was based on 87 personal interviews with health and social professionals from sectors such as NGOs, social services, trade unions, local institutions, and health services to promote compliance with these measures in four Spanish regions. It explored the difficulties faced by migrant agricultural workers in complying with prevention measures during the COVID-19 pandemic, the measures taken by organizations and public institutions and the health consequences of the pandemic on this group. After a reflexive thematic analysis, the results show…

Title: informative, does the term institutional responses encompass organizations such as NGOs?

Thank you for pointing out this remark. Yes, we were referring to both social and institutional (governmental) entities, so to make the title more inclusive, we modified it as follows: 

“Lessons learned from cross-sectoral collaboration to protect Migrant Farmworkers during COVID-19 in Spain”

Introduction: broad and well presented. Authors describe the study problem well. Well justified and contextualized study.

The words of the reviewer are highly appreciated

Methods: a qualitative descriptive study is an appropriate design for this work. Justify why the study was carried out in those 4 regions and not in others, would the conditions have changed? This may be a limitation of the study.

These regions were selected because they are the ones hosting the highest numbers of Migrant Farmworkers, in comparison with other regions in Spain. 

It has been specified like this at the end of the introduction sections: (line 169-170)

…in four Spanish regions that host the high numbers of MSWs (Catalonia, La Rioja, Murcia, and Andalusia)

What were the sample selection criteria? Although grouped sociodemographic data are given, inclusion/exclusion criteria are not clear. Define members of the sample well. Who are Members of associations/activists? Who are social integrators? Participation of all these groups in the sample should be better justified by explaining their functions in reference to these migrant workers. Is it purposive or convenience sampling? 

Thank you for pointing out this missing or confusing information. First, we have added a new sentence in the methods section explaining that the sampling strategy was purposive (since participants were selected based on their capacity to provide valuable information about the study objectives) and the inclusion criteria (health and social professionals whose tasks involved assisting migrant agricultural workers, for at least 6 months). (line 191-198)

The interviews involved professionals from public and social entities assisting MAWs: NGO staff, health workers, social workers and educators, intercultural mediators, agricultural trade unions, and people with managerial positions in public institutions (Table 1). Participants were selected by purposive sampling and met the following inclusion criteria: professionals working in public and social entities in the four above mentioned regions, whose professional tasks included assisting MAWs, and had a minimum of six months' experience working with them.

Second, in Table 1 we unified some similar profiles under generic classifications, since it’s true the activists and member of the associations were included in the NGOs, and the figure of social integrator is similar to a social worker. 

We have also removed a participant because the interview was held in Madrid, out of the regions included in the study (even though the person have worked in those regions). (line 202)

Table 1. Socio-demographic characteristics of the participants (n=92)

Sex

Men 38 (41%)

Women 54 (59%)

Professional profile 

Health professionals 17 (18%)

Social workers 24 (26%)

Cultural Mediators 8 (9%)

Members of NGOs 18 (19%)

COVID-19 Trackers 2(2%)

Trade unionist/worker representatives 14 (15%)

Political officers 8 (9%)

Businessmen 2 (2%)

Province

Almería 16 (17%)

La Rioja 20 (22%)

Lleida 24 (26%)

Huelva 11 (12%)

Jaén 4 (4%)

Murcia 17 (18%)

Age

18-25 1 (1%)

26-40 30 (33%)

41-50 36 (39%)

51-65 25 (27%)

Braum & Clark's thematic analysis phase could have been used as it was a qualitative/descriptive design. Incorporate an example of the coding process. 

We appreciate this suggestion. We have added an example of the coding process in a new figure 1. Find the fixed version on the picture below. 

Authors present approval from the Ethics Committee for Research with Medicines of the University Hospital Arnau de Vilanova-Lleida Territorial Management (CEIC-157 2459); This is a local committee, does this authorize them to obtain data in other regions of Spain? 

Having approval of the ethical committee allows gathering data under the conditions declared and approved by the committee, without restrictions related to the region of the participants. 

Who designed the interview script? 

The interview script was designed by the research team, including the topics that were of interest of the project, contrasted with literature. After its creation, it was piloted with 6 of the participants in the Andalusian region. After the piloting, the script was adjusted and completed with relevant topics that emerged during the interviews, and that were not included at the first version of the script. 

Who made the transcriptions of the interviews?

The audios were uploaded to the software NVIVO transcription, which produced the first written version of them. However, due to several mistakes and misinterpretations of the audio record, they were reviewed then one by one by the different members of the team. 

The study does not make clear whether data saturation was reached. 

At the same time that the data collection took place, some members of the research team where codifying the texts. Only when data saturation was reached in main topics as health, healthcare access living and labor conditions… and in all the regions, the data collection stopped. 

Well-developed rigor section. Use of COREQ checklist is important in these studies. 

 Following the suggestion of other reviewer, we moved the statement about the application of COREQ checklist to the rigor section. (line 227-228)

Results: interesting, very explanatory, they broadly reflect the response to the stated objectives. The names of the categories are not very risky, it seems that the analysis was deductive. 

We agree that the titles of the themes were so descriptive, despite we followed an inductive approach. We improved the names of the themes and subthemes by rephrasing them in a more interpretative way. You can check the new names of the themes along the results section and in the table 2 (line 262)

Table 2. Themes, subthemes, categories and subcategories

THEME SUBTHEME CATEGORY SUBCATEGORY

1. COVID's extraordinary experience of cross-sectoral collaboration 1.1. Convergence and Synergy in measures taken by regional and local authorities 1.1.1. The priority was that MSW were not dispersed and had accommodation

1.1.2. Accommodation for isolation in case of positive or close contact 

 1.2. Solidarity Initiatives taken by NGOs during the pandemic: prevention, food and hygiene 1.2.1. Health education 

1.2.2. Food and hygiene 

 1.3. Health system efforts to screen and vaccinate MWS 1.3.1. Screening strategies 

1.3.2. Vaccination strategies 

 1.4. Lessons learned and good practices of cross-sectoral collaboration and networking 

2. Compliance or non-compliance of MSW with the preventive and containment measures of COVID-19? 2.1. Challenges in practice to implement the prevention measures 2.1.1. Maintaining the safety distance between MSW At work

In accommodation

 2.1.2. Hygiene, disinfection and MSW’s use of protective face masks At work

In settlements 

 2.1.3. Performance of screening tests 

 PCR testing among MSW

Communication of PCR results to MSW

Registration of PCR results from MSW

 2.2. Grounds for MSW non-compliance with the preventive measures 2.2.1. MSW concealed symptoms for fear of losing work time and/or being evicted from the home 

 2.2.2. Cultural barriers and different conceptions of health/disease between MSW and the native population 

3. COVID exacerbated common health and healthcare problems of MWS 

 3.1. Imprints on physical and mental health of MSW 3.1.1. Physical health 

3.1.2. Mental health 

 3.2. The Odyssey of Care for MSW: Access and quality of healthcare 3.2.1. Neglect of other diseases 

 3.2.2.Telemedicine: increasing communication barriers and decreasing trust in the health system 

 Language barriers and diminished confidence 

Increasing communication barriers 

Discussion: I miss greater confrontation with results from similar studies in other European regions highlighting similarities/differences.

Thank you for the suggestion; some key elements have been added to help contrast our findings from a European perspective. See the added text at the discussion section:

Thus, the COVID-19 crisis has exposed, from a European perspective, the deep-rooted structural problem in the continent's agro-industry, especially in the south, where migrant labor operates under precarious conditions (45). In 2020, the European Commission issued guidelines to facilitate the mobility of workers deemed essential (54). Additionally, the New Pact on Migration and Asylum addressed migration as a key issue for the EU and offered a common framework for managing migration and asylum based on solidarity among all member states (55). These initiatives focused more on managing irregular migration and prioritizing market demands than on improving rights and safeguarding the health of migrants, specifically MFWs (46). In countries like Italy, actions were limited to the regularization of irregular migrant workers and the implementation of a “health pass” as a requirement for employment (56). This resulted in new roles and responsibilities for migrant workers, paving the way for the regularization of their status (57), but without substantial improvements in their social and working conditions in the short or medium term (56). On the other hand, Germany and Austria opted to allow the employment of migrants in agriculture and care, exempting them from travel bans, measures criticized for not ensuring adequate health protections for workers and being more oriented towards ensuring food availability for the entire population (58,59). In the UK, following the lockdown, agricultural organizations voiced concerns abou

---

## [Decision Letter · Decision Letter 1]

9 Jul 2024

Lessons learned from cross-sectoral collaboration to protect Migrant Farmworkers during COVID-19 in Spain

PONE-D-24-02040R1

Dear Dr. Perez-Urdiales,

We’re pleased to inform you that your manuscript has been judged scientifically suitable for publication and will be formally accepted for publication once it meets all outstanding technical requirements.

Kind regards,

Sanjit Sarkar, PhD

Guest Editor

PLOS ONE

Additional Editor Comments (optional):

Reviewers' comments:

Reviewer's Responses to Questions

**Comments to the Author**

1. If the authors have adequately addressed your comments raised in a previous round of review and you feel that this manuscript is now acceptable for publication, you may indicate that here to bypass the “Comments to the Author” section, enter your conflict of interest statement in the “Confidential to Editor” section, and submit your "Accept" recommendation.

Reviewer #1: All comments have been addressed

Reviewer #2: All comments have been addressed

Reviewer #4: All comments have been addressed

2. Is the manuscript technically sound, and do the data support the conclusions?

Reviewer #1: Yes

Reviewer #2: Yes

Reviewer #4: Yes

3. Has the statistical analysis been performed appropriately and rigorously? 

Reviewer #1: N/A

Reviewer #2: N/A

Reviewer #4: N/A

4. Have the authors made all data underlying the findings in their manuscript fully available?

Reviewer #1: Yes

Reviewer #2: (No Response)

Reviewer #4: No

5. Is the manuscript presented in an intelligible fashion and written in standard English?

Reviewer #1: Yes

Reviewer #2: Yes

Reviewer #4: Yes

6. Review Comments to the Author

Reviewer #1: The authors have done a good job of reviewing, they have responded appropriately to the questions raised and they have incorporated some suggestions made into their study. The study has improved considerably (especially in reformulating emerging themes from a more inductive perspective).

This is a good work on a very topical topic.

I have no more questions.

Reviewer #2: Dear editor

Thank you for sending me this interesting study for review.

The authors have adequately responded to almost all of the questions raised in my first review.

After having been thoroughly reviewed (6 reviewers), the authors have substantially improved the article.

Two minor issues could be improved in my opinion:

Data Analysis Section

AAlthough the authors now cite Braun and Clarke's thematic analysis procedure (50), the section has not really changed and does not describe the 6 steps proposed by Braun and Clarke: 1) data familiarisation, 2) systematic data coding, 3) generating initial themes, 4) developing and reviewing themess, 5) developing and reviewing themes, 6) writing the report.

Results

The inclusion of Table 2 (analysis results) greatly clarifies the results. In the naming of the analytical entities developed, the authors mix Themes/subthemes with Categories/subcategories. I suggest unifying the terminology and not mixing both concepts. Using Theme is consistent with Theme Analysis(ej. Main Theme, Theme, Subtheme, code)

Please see:

https://doi.org/10.1177/1049732308314930

https://doi.org/10.1016/j.ijnurstu.2020.103632

Reviewer #4: Lessons learned from institutional responses to protect Migrant Farmworkers during

COVID-19 in Spain

This paper examines the response of support staff, NGOs, service providers and various state actors to migrant agricultural workers in Spain during the COVID 19 pandemic. The paper employs interviews with the above participants to address the initiatives undertaken to address migrant workers health, to minimize their risk and exposure to COVID19 and the challenges in implementing these initiatives. The paper demonstrates how the structure of the program, irregular migration, the pre-existing precarity of MAWs, including their temporariness and mobility limited efforts to mitigate infection and provide care for MAWS. It also describes how coordinated efforts by a range of service providers fostered support and care for MAWs during COVID19, thereby laying groundwork for recommendations.

This second version of the paper is much improved. I am satisfied that the authors have addressed reviewers’ comments and provided much needed context (Europe, migration in Spain) as well as data analysis procedures, to make this paper worthy of publication.

Firstly, the paper clearly frames the issues in the context of the agricultural sector and the reliance of MAWs in Spain, including key elements of supports and the nature of workers precarity pre-COVID19. It justifies the regional focus in a convincing way and provides some detail on the provenance, diversity and cultural background of MAWs. The paper also situates the case of MAWs in Spain more explicitly in the European context. This makes the themes as discussed more convincing and relevant and it also makes the discussion and conclusions more convincing. The data analysis procedures are laid out in a convincing and detailed manner to make the themes that emerged from this inductive analysis, convincing.

The topical themes, however, provide evidence that suggest how employers/growers are implicated in the care or lack thereof for workers; this is theme worthy of greater attention considering workers fears of deportation and examples where adequate protection was not de facto provided. The paper also identifies changes to Spanish support policies that could facilitate improved care and inclusion of MAWs. The implications section is however, quite brief, making only 2 points (cultural mediator and need for collaboration). As the discussion suggests several factors related to training of service providers for example, could serve to strengthen this last section particularly since this is a descriptive rather than analytical paper.

Additional Note: it is not clear to me how authors have responded to the request for Data.

7. PLOS authors have the option to publish the peer review history of their article (what does this mean?). If published, this will include your full peer review and any attached files.

Reviewer #1: **Yes: **JOSÉ GRANERO MOLINA

Reviewer #2: **Yes: **Cayetano Fernández-Sola

Reviewer #4: No

---

## [Editor Report · Acceptance letter]

8 Aug 2024

PONE-D-24-02040R1 

PLOS ONE

Dear Dr. Perez-Urdiales, 

I'm pleased to inform you that your manuscript has been deemed suitable for publication in PLOS ONE. Congratulations! Your manuscript is now being handed over to our production team.

Kind regards, 

on behalf of

Dr. Sanjit Sarkar 

Guest Editor

PLOS ONE